# Strategies for Confronting the COVID-19 Pandemic in the State of Piauí—Brazil: Contributions to Nursing

**DOI:** 10.3390/ijerph21101384

**Published:** 2024-10-19

**Authors:** Thais Alexandre de Oliveira, Flor Marlene Luna Victoria Mori, Aracely Diaz Oviedo, Telma Maria Evangelista de Araújo, Daniela Reis Joaquim de Freitas, Andréia Rodrigues Moura da Costa Valle, Odinéa Maria Amorim Batista, Maria Zelia de Araujo Madeira, Neris Violeta González Pérez, Maria Eliete Batista Moura

**Affiliations:** 1Program in Nursing (PPGENF), Federal University of Piauí (UFPI), Teresina 64049-55, Brazil; thais.de@ufpi.edu.br (T.A.d.O.); telmaevangelista@ufpi.edu.br (T.M.E.d.A.); danielarjfreitas@ufpi.edu.br (D.R.J.d.F.); andreiarmcvalle@ufpi.edu.br (A.R.M.d.C.V.); oenf@ufpi.edu.br (O.M.A.B.); zeliamadeira15@ufpi.edu.br (M.Z.d.A.M.); 2Program in Nursing, Universidad Nacional de Trujillo (UNT), Trujillo 13000, Peru; flunavictoria@unitru.edu.pe; 3Program in Nursing, Universidad Autónoma de San Luis Potosí (UASLP), San Luis Potosí 78290, Mexico; aracelydiaz@uaslp.mx; 4Program in Nursing, Universidad de la República Oriental del Uruguay, Udelar, Montevidéu 11200, Uruguay; neris.perez@ufpi.edu.br

**Keywords:** coping strategies, COVID-19, public health, nursing

## Abstract

The COVID-19 pandemic has led to the adoption of rapid, complex, and changeable measures. These measures allowed effective care planning and implementation of emergency management strategies to meet the needs of the population. Objective: To analyze the influence of the strategies to cope with the COVID-19 pandemic, implemented by the Public Health Network of the State of Piauí, Brazil, as contributions to nursing, on the outcome of cases and deaths. Method: This is a descriptive and documentary study with a qualitative approach, carried out in the state of Piauí—Brazil, whose data were processed at IRAMUTEC and analyzed by the Descending Hierarchical Classification (DHC). Results: Ninety-two normative acts were instituted as coping strategies, distributed in three classes: technical-operational protocols for preventing and combating COVID-19 in Piauí; Piauí strategic bases for social distancing against COVID-19; and management of services and economic activities in Piauí in the fight against COVID-19. Conclusions: The actions determined by and implemented in the normative acts were conducted according to the behavior of the epidemiological curve regarding the number of cases and deaths. The normative acts, which defined the technical-operational protocols for the prevention and control of COVID-19, were directly related to social distancing strategies and the use of protective equipment in the quality of life of the population.

## 1. Introduction

In December 2019, in the city of Wuhan, in Hubei, China, the first case of Coronavirus disease, COVID-19, emerged (Coronavirus Disease, 2019), with sustained and exponential community transmission, emerging from an outbreak until it was declared a pandemic in March 2020 by the World Health Organization (WHO) [1].

Data showed more than 481,756,671 cases of COVID-19 worldwide, resulting in more than 6,127,981 deaths and more than 150,432,900 infections in the Americas during 2022 [2]. The first case of COVID-19 in Brazil was confirmed on 2 February 2020, and as of week 11 of 2022, 29,617,266 cases had been confirmed, with 657,102 deaths and 28,183,864 recoveries; in the state of Piauí, until the aforementioned epidemiological week, 367,193 cases had been confirmed, with 7714 deaths [3].

About 80% of the population recovered from the disease without hospital therapy, and one in six people infected with COVID-19 became seriously ill and developed difficulty breathing [4]. Most infected individuals developed mild symptoms, but about 14% required oxygen therapy due to worsening symptoms, and 5% required treatment at the intensive care unit, possibly requiring mechanical ventilation [5].

In the State of Piauí—Brazil, the Commission for the Control of Infections Related to Healthcare—CCIRAS and the Hospital Epidemiology Center—NHE, of the healthcare units in Teresina, the state capital, through emergency planning and training, highlighted the relevance of integration between healthcare teams and institutions for the continuity or redefinition of projects and actions, to guarantee universality, equity, and comprehensiveness of healthcare.

Nursing professionals, when leading the care provided to the population, felt the need to deepen their understanding of the relationship between the biological, the individual, the social, the political, and the concept of health-disease as a vital process and maintained their theoretical and practical considerations of humanizing care, combined with the individual, family, and social responsibilities that the pandemic required [6].

Considering the above, the study aims to analyze the influence of strategies aimed at fighting the COVID-19 pandemic, implemented by the Public Health Network of the state of Piauí, Brazil, as contributions to nursing, on the outcome of cases and deaths.

## 2. Materials and Methods

This is a descriptive and documentary study with a qualitative approach, carried out in the state of Piauí, in the Northeast region of Brazil. The state of Piauí has a public and private healthcare network consisting of 2093 healthcare establishments, of which 1498 are public (1440 municipal, 54 state, and 4 federal), and 595 are private (564 for-profit, 31 non-profit, and 293 in partnership with the Unified Health System-SUS) [7].

The data were collected from August 2022 to September 2023, in the databases of the state government, with authorization from the Health Department of the state of Piauí (SESAPI), from the notification forms of Influenza Syndrome (SG), in the E-SUS notification system, and Severe Acute Respiratory Syndrome (SRAG), in the SIVEP-GRIPE system. Regarding cases and deaths, public domain data were collected from the declarations available in the Mortality Information System (SIM), processed by the SUS Informatics Department—DATASUS of the Brazilian Ministry of Health [8].

The data on the coping strategies adopted in the period from 2020 to 2022 were obtained through 92 normative acts published on the SESAPI website (opinions, ordinances, decrees, laws, and provisional measures).

The inclusion criteria adopted were documents officially published on the subject and available on the official website of the government of the state of Piauí. Documents not published in the Official Gazette of Brazil and official communication channels on the SESAPI website were excluded.

The data contained in the SESAPI normative acts were processed using the IRAMUTEQ^®^ software (Interface de R pour les Analyses Multidimensionnelles de Textes et de Questionnaires) [9], version 0.7 alpha 2, and analyzed using the Descending Hierarchical Classification (CHD). Since the research was based on the texts of each Normative Act, each one comprised an Initial Context Unit (ICU), whose topic is unique and central, and after the program recognized the initial units, they were divided into elementary context units (ECUs).

For the CHD of the data, the contents of the published documents were formatted into a single text, called corpus, identified by coded variables characterizing the normative acts (act number, type, year, and month). The ICUs were reformatted into ECUs, identifying the frequency of these words in percentage and chi-square (χ^2^) or significance level and dividing common words into interconnected classes to be put into discussion.

The Normative Act variable (act) was coded from 1 to 92, respecting the order of time in which the documents were published; the type variable (tip) was coded from 1 to 5, for opinion, decree, law, ordinance, and provisional measure, respectively. The coding for year (anno) was from 1 to 3, being 1 for the year 2020, 2 for the year 2021, and 3 for the year 2022. In addition, the coding for month (month) was given in a window from 1 to 12, following the exact sequence of the months of the year, from January to December.

Regarding ethical aspects, Ordinance 466/12 of the National Health Council of the Brazilian Ministry of Health, which regulates research involving human beings, was considered [10]. This research was assessed by the State Coordination of Epidemiology/CIEVS of SESAPI and then provided with the necessary documentation, approved by the Research Ethics Committee of the Federal University of Piauí, through the Certificate of Presentation for Ethical Assessment—CAAE N. 54371521.9.0000.5214.

## 3. Results and Discussion

### 3.1. Representation of Cases and Deaths from COVID-19 in Piauí, Brazil

In the state of Piauí, 297,208 cases of COVID-19 were recorded between 2020 and 2022, with a prevalence of 28.64 cases per one thousand inhabitants in 2020, 27.68 in 2021, and 34.31 in 2022, exceeding the cases in 2020.

The [Fig ijerph-21-01384-ch001] below shows the behavior of cases in the analyzed years.

At the beginning of the pandemic, the state of Piauí stood out for having the highest fatality rate in the northeast region and the lowest number of confirmed cases. However, there were no official data on the exact number of tests performed to diagnose COVID-19, implying that the fatality rate may not have represented the real situation of the epidemic in the state [11]. In this sense, COVID-19 cases in the state of Minas Gerais were identified through testing only of severe cases, leading to possible underreporting [12], with testing limited to symptomatic patients and household contacts of confirmed cases [13].

Among the COVID-19 mortality rates analyzed in the period from 2020 to 2022, there were 8324 deaths, with the highest rate in 2021, with 1.4 deaths per 100,000 inhabitants. [Fig ijerph-21-01384-ch002] below shows the behavior of mortality during the research period, revealing a decrease after the peak of 2021.

### 3.2. Characterization of Normative Acts to Fight COVID-19 in the State of Piauí

To characterize the normative acts of the study, the following variables were established: type of act, year of publication, and month of publication, as shown in Table 1.

Of the 92 normative acts published between 2020 and 2022, 1 was an opinion (1.1%); 86 were decrees (93.5%); 2 were laws (2.2%); 2 were ordinances (2.2%); and 1 was a provisional measure (1.1%). Regarding the year variable, 47 normative acts were published in 2020 (51.1%), of which 44 were decrees (93.6%), 2 were ordinances (4.3%), and 1 was a provisional measure (2.1%); in 2021, 43 normative acts were published, of which 41 were decrees (95.4%) and 2 were laws (4.6%); and up to March 2022, 2 normative acts (2.2%) were published, of which 1 was a decree (50.0%) and 1 was an opinion (50.0%).

According to the month variable, in the period from 2020 to 2022, 1 normative act was published in the months of January (1.1%); 3 in February (3.3%); 13 in March (14.1%); 11 in April (12.0%); 7 in May (7.6%); 19 in June (20.7%); 14 in July (15.2%); 7 in August (7.6%); 6 in September (6.5%); 6 in October (6.5%); 2 in November (2.2%), and 3 in December (3.3%), with the largest number of publications still in 2020, the first year of the pandemic.

The distribution of variables in Class 1 shows year 1 (2020) as the most significant (χ^2^ 316.62), followed by normative act 1, which, due to the urgent nature of the situation imposed on the state, required clearer and more comprehensive technical-operational protocols. In Class 2, year 2 (2021), the normative acts of the first half of the year stood out, in which there was still an increase in the number of cases and deaths from COVID-19. Class 3 revealed year 2 (2021) with the greatest significance (χ^2^ 150.44), followed by the last normative acts evaluated, in which there was greater flexibility in strategic coping actions (Figure 1).

The IRAMUTEQ^®^ software version 0.7 divided the study corpus into 1577 ECUs and 1211 segments and three semantic classes, with 76.79% of the material being used. Class 1: Technical-operational protocols for preventing and combating COVID-19 in Piauí (46.16%); Class 2: Strategic bases of Piauí for social distancing against COVID-19 (21.64%); and Class 3: Management of Piauí services and economic activities in the fight against COVID-19 (32.2%) (Figure 2).

The three semantic classes allowed analyzing the strategies for dealing with the COVID-19 pandemic in the state of Piauí and the distribution of words in percentage and chi-square (χ^2^), represented in Figure 2, analyzed from left to right, as guided by the software.

### 3.3. Class 1—Technical-Operational Protocols to Prevent and Fight COVID-19 in Piauí

Consisting of 559 text segments and 46.16% of the ECUs in the category, class 1 is directly related to class 2 and associated with class 3. In this class, words such as measure, sanitary, health, protocol, hygienic-sanitary, recommendation, determined, and containment represent the technical-operational protocols for preventing and fighting against COVID-19 in Piauí and show that the strategies adopted by the state of Piauí were, for the most part, aimed at direct actions in individual and collective health care, with instructions ranging from correct hand hygiene, environment hygiene, and protective equipment, considering the national measures established together with the WHO.

Hygienic and sanitary practices were established in accordance with WHO guidelines over time and reproduced under guidelines in the following normative acts:

*[…] Art. 13 recommends that private establishments and public agencies adopt the following sanitary measures: i—provide places to wash hands frequently; ii—provide dispensers with 70% alcohol gel; iii—provide disposable paper towels; iv—increase the frequency of cleaning floors, handrails, door handles and bathrooms with 70% alcohol or bleach solution […]*.(Act 1)

*[…] Art. 1 this decree provides for the mandatory use of a face mask as an additional measure necessary to combat COVID-19. Art. 2 determines the use of face masks made according to the guidelines of the Ministry of Health (…) 1 it will be mandatory to wear a face mask whenever it is necessary to leave the house, travel on public roads or remain in spaces where other people circulate […]*.(Act 9)

Still in February 2020, SESAPI implemented the State Contingency Plan to fight human infection by the Coronavirus (2019-nCoV) in the state of Piauí, so that health professionals would have rapid updates and access to the conducts and protocols to be followed. This plan was based on three levels of response: alert, imminent danger, and public health emergency, with each level based on the assessment of the risk of the new Coronavirus affecting the country and its impact on public health [14].

The state contingency plan took into account the transmissibility, pathogenicity, and virulence of COVID-19, as well as its geographic spread, population vulnerability, and preventive measures, such as vaccines and specific treatments, in addition to general WHO recommendations and scientific evidence. For that purpose, it re-implemented the Strategic Health Surveillance Information Center (CIEVS, Centro de Informações Estratégicas de Vigilância em Saúde) to ensure support for all notifiable diseases, outbreaks, epidemics, and pandemics [14].

Given the absence of a vaccine and specific antiviral medication, behavioral measures are constantly reinforced as the main tools to fight COVID-19. Therefore, the use of facial masks, respiratory etiquette, as well as social distancing, early diagnosis with case isolation, and quarantine of contacts were crucial in controlling the transmissibility of the virus and mitigating the social, economic, and epidemiological risks generated by the disease [15].

As essential members in the planning, implementation, and evaluation of care, nursing professionals were subject to weaknesses in the COVID-19 pandemic due to the lack of organization and adequate protocols, inequality in the distribution of personal protective equipment—PPE—regarding quantity and quality, and lack of training to use them, a fact that requires the development of strategies for reorganizing care and improving working conditions [16,17]. Moreover, there was a lack of diagnostic tests and technical-operational preparation, especially in public institutions, in which PPE was re-sterilized and/or reused, hindering the access of these professionals to some health service environments due to the lack of protective equipment, resulting in unstable or even unsafe patient care due to the shortage of nursing personnel [18].

It is imperative to implement a contingency plan that encompasses the entire management of human, physical, and material resources with the representative entities in the institution or municipality to cover the maximum service needs and guarantee the surveillance and management of confirmed cases of patients and professionals, flow of care, screening methods, training, distribution of PPE, team awareness, and monitoring of supplies, among others [19].

As of June 2020, normative acts in Piauí started to deal with the process of gradual resumption of activities in the state through the organized resumption pact, Pró-Piauí, established by Decree N. 19.014/2020, which provided for the rules of transition necessary to make health measures more flexible [20]. This flexibility was inversely proportional to the transmissibility of the disease and the occupancy rate of ICU beds in the state.

*[…] approves specific protocols with measures to prevent and control the spread of SARS-CoV 2/COVID-19 for the construction chain and provides other guidance for employers, workers and customers in the construction chain, including retail and wholesale trade of the sector […]*.(Act 28)

*[…] Art. 1—the specific protocol with measures to prevent and control the spread of SARS-CoV 2/COVID-19 for the education sector is approved in the form of the sole annex to this decree […]*.(Act 40)

In parallel, a commission was established to plan the selection process, aiming at the temporary hiring of health professionals to meet the urgent demands for labor in health units in the state of Piauí [21].

Criteria were also established for reopening all activities:

*[…] 1—if the protocols and sanitary measures to fight COVID-19 are followed, social cultural and artistic sporting activities and events may be carried out with the following public metric and immunization restrictions (…); ix—the vaccination to be proven must correspond to at least 2 two doses or a single dose of vaccines against SARS-CoV-2 […]*.(Act 88)

At the end of 2021, with the creation of vaccines against COVID-19, proof of vaccination was established as one of the criteria for increasing the flexibility of economic and social activities in the state of Piauí. Better known as the ‘vaccination passport’, the presentation of the vaccination card now gives citizens of Piauí the right access to work and to the most diverse public environments.

*[…] Art. 1—(…) a vaccination passport against COVID-19 will be required for the purposes of access to in-person service in public administration bodies and entities. 1—the vaccination to be proven must correspond to minimum of 2 two doses or single dose of vaccines against SARSCOV-2 according to the schedule established by the municipal health departments in relation to the individual’s age from 18 years of age. 2—the vaccination passport will be required from public servants and employees. 3—in addition to the corresponding disciplinary measures, the employee will lose their remuneration for the days they are absent from work for not presenting their vaccination passport […]*.(Act 89)

However, the hygiene recommendations previously established in the normative acts mentioned before were maintained for the entire state.

*[…] on that occasion, the members of COEPI unanimously expressed to be in favor of the following recommendations to the Health Secretariat of the state of Piauí, other health authorities and the state government: 1—maintain mandatory use of masks in closed public or private spaces; 2—provide the use of masks in open and semi-open spaces with the recommendation of mandatory immunization with the booster dose (3rd dose), in accordance with the vaccination schedule […]*.(Act 92)

The creation of the nursing Committee to fight COVID-19 in Bahia, Brazil, helped to guide, support, and act in the defense of nursing, receiving demands, doubts, and questions from workers through communication, review of contingency plans, advice to institutions, epidemiology, and external activities in favor of safe care and monitoring of health and safety conditions at work. Thus, it was able to indicate ways to overcome the regulation of wages, working hours, decent rest, and respect for staffing [22], allowing the interconnection of social actors to reach the political agenda in the treatment and solution of problems.

### 3.4. Class 2—Strategic Bases of Piauí for Social Distancing Against COVID-19

Consisting of 262 text segments and 21.64% of ECUs in the category, class 2 is directly related to class 1 and also associated with class 3, representing the strategic bases of Piauí for social distancing against COVID-19. In this class, the words *circulation*, *person*, *need*, *understand*, *displacement*, *legislation*, *direction*, *effect*, and *prohibition* reveal the state intervention regarding the movement of people and contacts with other individuals in the most diverse environments, aiming to avoid possible contamination and the spread of the new virus.

*[…] 4—in hotels, meals will be provided exclusively through room service. 5—in establishments and activities in operation, it is mandatory to control the flow of people to prevent crowds (…) 1—the suspension of religious in-person activities in churches or temples is determined. 2—the suspension of activities in parks or other spaces accessible to the public that encourage crowds […] is determined*.(Act 4)

Measures were also established regarding the control of the flow of people within the state of Piauí and for those who enter the state by air, rail, road, and sea transport. In the first case, control was carried out by state health surveillance services, together with municipal and federal health surveillance, and with support from the military police, civil police, federal police, and federal highway police [23]. Regarding travel by public transport, a minimum quarantine of 7 (seven) days was established under complementary SESAPI action rules [24].

With the exponential increase in the number of cases and deaths in Piauí due to COVID-19, the government launches normative acts with social isolation measures to be adopted throughout the state:

*[…] only the following essential activities and establishments may operate: i—pharmacies and drugstores; ii—health services; iii—supermarkets; iv—bakeries and bakeshops; v—gas stations; vi—tire shops; vii—delivery services; viii—security and surveillance services; ix—telecommunication services, broadcasting and press; x—banking services exclusively for the paying of emergency aid and social benefits and for self-service […]*.(Act 11)

It was verified that specific spaces and certain forms of social interactions contribute to the spread of COVID-19, such as: transmission in households, with the mobility of family members; schools and universities, due to the mobility and concentration of social groups; in workplaces, with precarious PPE and high risk activities; as well as in public places and communities (stadiums, markets, squares, and organized tours), where the spread of the disease can reach an extensive network of people of different age groups [25].

The fight against COVID-19 overlapped with the reality of people, health managers, and workers, who suffer daily with negative feelings about the current situation, in addition to often canceling themselves as people in their basic human needs in favor of professional practice. Fear, anxiety, loneliness, sadness, physical and mental fatigue, lack of respect, abandonment, and frustration were prominent feelings [16].

Irritability, anxiety, and uneasiness have already been demonstrated by health professionals, whose concern was not being infected but infecting their families, in addition to the lack of cooperation from patients in maintaining precautionary measures and impotence in the face of severely ill patients, generating psychological disorders [26].

With PRO-Piauí, the easing of social isolation measures in public and private environments started, maintaining its progression or regression depending on the number of cases and deaths registered, as demonstrated in the following normative acts:

*[…] remain suspended: i—in-person service and consumption at the establishment of food and beverage services located on beaches, resorts, waterfalls, museums, libraries, zoos; ii—tourism services in parks, beaches, spas, waterfalls, museums, libraries and zoos; […]*.(Act 39)

*[…] in-person educational activities relating to: (…) a—minimum occupancy of 4 square meters per person may operate from September 22, 2020; b—minimum distance of 2 m between people (…) v—educational events such as lectures, symposiums and congresses (…) as long as they are held: a—in an open or semi-open environment with air circulation; b—with the presence of up to 100 hundred people. vi—dance and music schools, soccer schools, gyms, swimming schools […]*.(Act 40)

Studies of the impact of non-pharmacological measures revealed that the low number of cases in the pediatric population could be attributed not only to the non-detection of the disease due to mild or absent symptoms but also to the early closure of schools, which reduced the transmission of the virus [27].

The consumption of alcoholic beverages in public places and around bars and restaurants was still prohibited until the end of 2020, with the exception of cases of consumption inside the establishment, with customers properly seated in chairs or seated at tables, with a minimum distance of 2 m between them and respecting the established hygienic-sanitary protocols, in accordance with Decrees 19,278, 19,288 and 19,318 of 2020 [28,29,30].

In addition to the social distancing and isolation measures adopted by the government of the state of Piauí to contain the spread of COVID-19 in its management territory, other social coping strategies were highlighted, such as optional work days in linked public offices and transfers of holidays such as weekend extensions, aiming to reduce crowds and the stay in environments other than their homes:

*[…] Art. 1—the celebrations related to Public Servants’ Day celebrated annually on October 28 are transferred to October 30, 2020. Art. 2—an optional work day is decreed on October 30, 2020 in all bodies and entities of the direct and indirect autarchic and foundational state public administration of the executive government without harm to the essential services on which the holder of the bodies and entities will decide […]*.(Act 44)

Regarding the role of nursing when facing political, economic, and health challenges, care with technical and human competence is identified as a mediator between family members and hospitalized people, reaffirming their identity and ideas as a social practice to defend life, concrete conditions of production and reproduction, and universal access to health, aiming to achieve social justice [17].

In the beginning of 2021, activities involving gatherings and cultural events, as well as sporting activities and any spaces that promoted festive activities, whether public or private, indoors or outdoors, remained suspended:

*[…] ii—bars and restaurants will only be able to operate until 11 p.m. and the use of ambient sound, whether through live music or mechanical or instrumental sound, is prohibited; iii—commerce in general can only operate until 5 p.m. and shopping malls from 12 p.m. to 9 p.m. (…) v—the permanence of people in open public spaces for collective use such as parks, squares, beaches and others is subject to strict obedience to the health protocols of state and municipal health surveillance, especially regarding the mandatory use of masks […]*.(Act 48)

Around the second half of 2021, with the reduction in the incidence of cases and deaths, there was an increase in the flexibility regarding isolation and social distancing, opening space for cultural and artistic activities in cinemas, theaters, circuses, auditoriums, and events in open and semi-open spaces with a maximum audience of 200 (two hundred) people, with a minimum distance of two meters, and with the presence of mechanical instrumental sound or musicians, as long as it did not generate crowds or dancing. Prohibitions on gathering of people and consumption of alcoholic beverages in public places or on movement and driving under the influence of alcohol were maintained [31].

Decrees numbers 20,036 and 20,150 of 2021 brought even greater flexibility subsequently, increasing the number of people to 500 (five hundred) in open and semi-open spaces and 200 (two hundred) in closed spaces, already establishing 2 (two) doses of the immunizing agent, or a single dose, or negative tests (antigen or RT-PCR) 48 h before the event. In the case of shows, those with a seated audience, without dancing; theaters and cinemas with 50% capacity; soccer games, courts, with a capacity of 30% of the public and seated audience, with a distance of 1.5 m between people being established for all occasions [32,33].

With the consolidation of vaccination coverage and reduction in the number of cases and deaths, in the beginning of 2022, measures to make social activities more flexible with an audience of 100% (one hundred percent) of capacity were expanded, as long as general hygienic-sanitary measures were followed, such as distancing of 1.5 m and proof of immunization against COVID-19 in economic establishments and events, among the public, workers, and collaborators [34].

### 3.5. Class 3—Management of Services and Economic Activities in Piauí in the Fight Against COVID-19

Consisting of 390 text segments and 32.2% of ECUs in the category, class 3 is associated with classes 1 and 2, representing the management of services and economic activities in Piauí in the fight against COVID-19, as suggested by the words *open*, *environment*, *restaurant*, *bar*, *party*, *commerce*, *customer*, *closed*, *mechanic*, *shopping*, *operation*, *use*, and *sound*. Thus, the coping strategies adopted to fight COVID-19 in the state of Piauí were also aimed at the operating modes of commercial establishments and events in general that generated crowds and directly interfered in the disease transmissibility and increased the number of cases and deaths, making the health logistics difficult.

Regarding the means of transportation, there was an inter-municipal suspension in the transportation of passengers, with a fine being applied when there was an infraction, and a protocol related to the transport of cargo was under monitoring by the state health surveillance, together with municipal and federal entities, as well as the military police, civil police, federal police, and federal highway police [35]:

*[…] Art. 1 the following are suspended (…) intercity passenger transport services in road modality classified as service: i conventional; ii alternative; iii semi-urban; iv chartered (…). Failure to comply with the suspension determined by this decree will subject the offender to the penalty of vehicle retention without prejudice to the imposition of a fine or other sanction (…). […]*.(Act 7)

Regarding trade, suspension and subsequent flexibility of all economic activities in the state of Piauí were established at intervals of time according to the incidence of cases and deaths throughout the pandemic period and following the health determinations issued by SESAPI:

*[…] Art. 1—the suspension of all activities in bars, restaurants, cinemas, clubs, gyms, concert halls and beauty clinics is determined; public and private dental oral health activities, except those related to urgent and emergency care; sporting events; of commercial activities in shopping malls […]*.(Act 3)

During this period, some reservations were made regarding the aforementioned suspension, as long as compliance with individual protection was ensured, the flow of people was controlled to avoid crowds, and a plan was presented to reduce activities by 50%, including working hours and protective measures. for workers and management.

As of May 2020, the operation of some essential activities and services was determined, such as pharmacies, health services, supermarkets, bakeries, fuel stations, tire repair shops, delivery services, security and surveillance services, telecommunications services, broadcasting and press, as well as banking services, exclusively for the payment of emergency aid and social benefits and for self-service [33]. Agricultural and agro-industrial activities were also included, including harvesting, milking, storage, and drying, among other activities at risk of perishing [36].

Regarding events in general, collective activities or events held with more than 50 people were suspended, as well as sporting, artistic, cultural, political, scientific, commercial, religious, and other mass events [37]. In 2021, festive activities (including carnival and its previews), sports, cultural events, and the operation of nightclubs and concert halls remained suspended, whether in open or closed environments, public or private, with or without ticket sales:

*[…] iv—temples, churches, spiritualist centers and terreiros may operate under the restrictions of the specific health protocol for Holy Week established by the Piauí State Health Secretariat (…) 5 the use of beaches is prohibited, as well as resorts, waterfalls and parks […]*.(Act 58)

With social isolation measures in public spaces and events as the main means of protection against exposure to the virus, on the other hand, people with lower income, who live in unhealthy and crowded housing, had greater difficulty maintaining social isolation, not only due to the housing structure and the financial conditions to maintain food and cost balance, making equitable governmental social and financial support actions necessary [38].

With the advancement of the COVID-19 pandemic, which initially kept professionals from risk groups away, temporarily but progressively, as well as the absence of infected professionals, a shortage of labor in health services. Thus, crisis coping strategies needed to be established to maintain what is necessary to guarantee quality assistance and allow professionals who were exposed to the virus but who were asymptomatic to remain at work, testing them through RT-PCR, maintaining a normal temperature, and following strict hygiene guidelines and protection measures with the use of personal protective equipment (PPE) [39].

With high levels of economic crisis in the state and following the line of emergency aid implemented by the federal government, the government of the state of Piauí instituted a law granting emergency aid to individual microentrepreneurs—IME—and establishments opting for the simple national tax, including sectors of bars, restaurants, and event organizers, as well as workers in these sectors who lost their jobs and were left without any social security benefits or unemployment insurance [40].

In education, there was a suspension of classes in the state and municipal, public, and private networks, including higher education institutions, considered in the school calendar as an anticipation of the school holidays corresponding to the month of July, not applying to activities carried out with the use of electronic platforms, which do not require in-person activity [37,41].

Regarding the coping strategies of nursing professionals in higher education institutions, students in the last year attended the practice with maximum possible safety; the lines of research followed the WHO guidelines on COVID-19, and several related projects are presented in the extension, with the home activity regime for theoretical activities; extension activities for those who are not at risk; remote *stricto sensu* postgraduate studies; and active administrative staff, such as security measures, the use of alcohol gel dispensers, and professional training [42].

The organized resumption established greater management of economic activities both in their categories and in their opening hours, according to the observance of cases and deaths in the state and the presentation by the establishments of the health security plan. However, some environments and services remained suspended:

*[…] remain suspended: i—in-person service and consumption at the establishment of food and beverage services located on beaches, resorts, waterfalls, museums, libraries and zoos; ii—tourism services in parks, beaches, resorts, waterfalls, museums, libraries and zoos; iii—tourist services in municipalities located in health care regions focused on COVID-19 […]*.(Act 39)

The measures of the normative acts, considered universal and common to other Brazilian states, were able to contain the spread of the disease and its virulence at times. As the months went by, analyzing the behavior of the curve of cases and deaths from SARS-CoV-2, the measures and the outcome began to have an inverse relationship of influence because, as the number of cases and deaths in the state decreased, the measures regarding social isolation, reopening of businesses, and strictness of hygiene and sanitation actions were relaxed. Thus, over time, it was the cases and deaths that began to influence the strategies to combat the disease.

However, with the qualitative approach to data, the limitations of the study are related to the lack of correlation analysis with professional efficiency, population behavior, and public health interventions and the confirmation of causality, requiring further investigation to confirm causality. Another limitation of the study refers to the possibility that official documents and normative acts may produce biases if they do not fully capture local realities or if certain actions are not adequately documented.

## 4. Conclusions

In the state of Piauí—Brazil, 297,208 cases of COVID-19 were registered in the period from 2020 to 2022, which corresponds to an average prevalence of 30.21/1000 inhabitants and 8324 deaths, at an average crude rate of 0.8 deaths per 100 thousand inhabitants. The actions determined and implemented in the normative acts, with the participation of nursing, were valid for the emergency that the world was facing, being conducted through the behavior of the epidemiological curve regarding the number of cases and deaths.

The normative acts, which defined the technical-operational protocols for the prevention and control of COVID-19 in the state of Piauí, were directly related to social distancing strategies and the use of personal protective equipment by the population as more significant for the fight against COVID-19, determining the needs and costs for the management of health services and the development of economic activities during the pandemic.

The strategies were previously taken in accordance with the precepts of the World Health Organization (WHO) and the federal government as the standard, which may have compromised the expected result due to the low adaptation to the economic, social, and health care realities of the state. These strategies influenced the outcome of cases and deaths, being rigorously implemented or attenuated according to the behavior of the epidemiological curve.

The study addresses the expressions of individual and collective coping as well as the creation of structural, political, union, psychosocial, and professional organizations that place nursing at greater visibility in society. The use of mathematical and statistical models is fundamental for efficient management in several areas, as well as educational strategies to maintain the quality of education, training, the relationship with the environment, and other forms of care that meet the physical and spiritual aspects. However, care is the centrality of nursing’s work, essential to saving lives.

Thus, the study can contribute to improving the health care network and the quality of life of the population through adequate infrastructure, strategic planning of actions and evaluation of activities, use of specific materials and equipment, professional training, and readjustment of teams to new health demands. The intellectual and scientific contribution in the applicability of the proposed actions and stimulus to similar research and further studies on the consequences of the pandemic and its strategies on the health of the population as a whole, in helping other professionals and managers to perpetuate care, is also estimated.

## Data Availability

The entire data set that supports the results of this study was published in the article itself.

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
