# Peer review of "Strategies for Confronting the COVID-19 Pandemic in the State of Piauí—Brazil: Contributions to Nursing"

_ijerph, 2024, doi:10.3390/ijerph21101384_

Round 1
Reviewer 1 Report
Comments and Suggestions for Authors
Dear Authors,
Thank you for submitting your manuscript titled "Strategies for Coping with the COVID-19 Pandemic in the State of Piauí – Brazil: Nursing Contributions." Your work addresses a significant and timely topic, offering valuable insights into the regional strategies employed to combat the COVID-19 pandemic. However, after a thorough review, I have identified several areas where the manuscript can be strengthened to ensure it meets the highest standards of academic rigor and clarity.
1. Data Interpretation and Analysis:
Overestimation of Strategy Impact:
The manuscript asserts a strong and direct impact of the coping strategies on COVID-19 outcomes in Piauí. However, this conclusion appears to be based primarily on descriptive analysis without sufficient statistical rigor to establish causality. I recommend revisiting these claims and conducting more robust statistical analyses, such as multivariate regression or time-series analysis, to accurately assess the impact of the interventions. This would help avoid overestimating the effectiveness of the strategies without considering other influencing factors.
-
The manuscript could benefit from a more comparative approach, perhaps by comparing Piauí’s strategies and outcomes with those of other Brazilian states or countries with similar socioeconomic conditions. This would provide context and allow for a more nuanced understanding of the effectiveness of the strategies implemented.
-
While the descriptive nature of the study is valuable, the manuscript could be strengthened by incorporating more analytical elements. For instance, the impact of specific strategies on COVID-19 outcomes (cases, mortality, etc.) could be statistically analyzed to draw more definitive conclusions.
Correlation Misinterpretation:
The discussion on the correlation between the number of cases and deaths suggests a causal relationship that may not be fully substantiated. Correlation does not imply causation, and without controlling for potential confounding variables (e.g., healthcare capacity, public health interventions beyond those documented, population behavior), this interpretation could be misleading. I advise a more cautious presentation of these findings, perhaps by highlighting the limitations of correlation analysis and clarifying that the observed relationships require further investigation to confirm causality.
Underestimation of Confounding Factors:
The manuscript does not fully account for other factors that may have influenced the COVID-19 outcomes in Piauí, such as the role of healthcare infrastructure, public adherence to guidelines, and the timing and extent of vaccination efforts. These factors are crucial for a comprehensive analysis and should be incorporated into your discussion. Acknowledging these elements will provide a more balanced and accurate portrayal of the situation, thereby enhancing the validity of your conclusions.
Limited Discussion of Nursing’s Role:
Although the title of the manuscript suggests a focus on nursing contributions, the actual discussion of how nursing professionals specifically influenced the outcomes is somewhat limited. To align the content with the title, I recommend expanding this section to include detailed examples of nursing interventions and their impact. Highlighting the unique contributions of nursing, including leadership, patient care innovations, and collaboration with other healthcare sectors, would greatly enhance the manuscript’s relevance and depth.
The conclusions drawn in the manuscript tend to be general and could benefit from being more specific and actionable. I recommend offering detailed recommendations based on your findings, particularly regarding the ongoing role of nursing professionals in pandemic management and specific strategies for improving public health responses in similar contexts.
The study relies heavily on official documents and normative acts, which may introduce bias if these documents do not fully capture the on-the-ground realities or if certain actions were not adequately documented. A discussion on the potential limitations of using these sources would be beneficial. The references included in the manuscript primarily consist of governmental and official documents. While these sources are certainly valuable, the manuscript would benefit from the inclusion of more peer-reviewed academic articles, especially from journals in public health, epidemiology, and nursing. These sources would provide a stronger foundation for your arguments and contribute to the academic rigor of your study.
I look forward to seeing the revised version of your manuscript.
Best regards,
Comments on the Quality of English LanguageThe manuscript contains several run-on sentences that combine multiple ideas without proper punctuation, making them difficult to follow. For example:
"The COVID-19 pandemic determined the adoption of rapid, complex and changeable health action measures, allowing care planning and emergency management strategies to meet the needs of the population."
This sentence could be split into two sentences for clarity: "The COVID-19 pandemic necessitated the adoption of rapid, complex, and changeable health action measures. These measures allowed for effective care planning and the implementation of emergency management strategies to meet the population's needs."
Fragmented Sentences:
Some sentences lack a clear subject or verb, making them incomplete. For instance:
"Considering the above, the study aims to analyze the influence of strategies to fight the COVID-19 pandemic of the Public Health Network of the state of Piauí, Brazil, on the outcome of cases and deaths."
This sentence is long and convoluted. It could be restructured as: "This study aims to analyze the influence of COVID-19 pandemic strategies implemented by the Public Health Network in the state of Piauí, Brazil, on case and death outcomes."
2. Tense Consistency:
Mixed Tenses:
The manuscript frequently switches between past and present tenses, which can confuse the reader. For example:
"The data showed more than 481,756,671 cases of COVID-19 worldwide and more than 6,127,981 deaths, with more than 150,432,900 people infected in the Americas up to March 28, 2022."
It would be more consistent to use the past tense throughout: "The data showed more than 481,756,671 cases of COVID-19 worldwide, resulting in over 6,127,981 deaths, and more than 150,432,900 infections in the Americas as of March 28, 2022."
3. Subject-Verb Agreement:
Singular vs. Plural Mismatch:
There are instances where the subject and verb do not agree in number. For example:
"The data on COVID-19 cases was collected from SESAPI's information and statistics system."
"Data" is plural, so the correct form should be: "The data on COVID-19 cases were collected from SESAPI's information and statistics system."
4. Article Usage:
Missing Articles:
Articles such as "the," "a," and "an" are often missing, leading to awkward phrasing. For example:
"COVID-19 pandemic determined adoption of rapid, complex and changeable health action measures."
Should be: "The COVID-19 pandemic determined the adoption of rapid, complex, and changeable health action measures."
5. Word Choice and Redundancy:
Improper Word Choice:
In some cases, the wrong word is chosen, affecting the clarity of the sentence. For instance:
"The actions determined in the normative acts were valid for the emergency that the world was facing."
A more appropriate word choice could be: "The measures outlined in the normative acts were appropriate for the emergency the world was facing."
Redundancy:
Redundant phrases can make sentences wordy and less impactful. For example:
"Due to the lack of protective equipment, resulting in unstable or even unsafe patient care, due to the shortage of nursing personnel."
This can be streamlined to: "The lack of protective equipment resulted in unstable and unsafe patient care due to the nursing personnel shortage."
6. Punctuation Errors:
Comma Usage:
Commas are often misplaced or missing, which can alter the meaning of a sentence. For example:
"The role of nursing professionals through the care provided to the population is the subject of reflection and analysis indicating the need to deepen the understanding."
It should be: "The role of nursing professionals, through the care provided to the population, is the subject of reflection and analysis, indicating the need to deepen the understanding."
7. Prepositions:
Incorrect Prepositions:
Prepositions are occasionally misused, leading to awkward phrasing. For example:
"Increased transmissibility and virulence, associated with changes in the clinical presentation of the disease, low diagnostic efficacies and public health measures, as well as low availability of vaccines and therapies or even mutations in the virus itself led to the emergence of SARS-CoV-2 variants."
A clearer phrasing would be: "Increased transmissibility and virulence, combined with changes in the clinical presentation of the disease, low diagnostic efficacy, public health measures, and the limited availability of vaccines and therapies, or even mutations in the virus itself, led to the emergence of SARS-CoV-2 variants."
8. Spelling Mistakes:
Typographical Errors:
There are a few spelling errors or typos that need correction. For instance:
"Public Health Netwok" should be corrected to "Public Health Network."
9. CClarity and Conciseness:
Ambiguity:
Some sentences are ambiguous and need to be rephrased for clarity. For example:
"The strategies adopted in Piauí were essential to save lives, with other actions being approved or blocked depending on the behavior of the epidemiological curve."
This could be clearer as: "The strategies adopted in Piauí were essential in saving lives. Other actions were either approved or blocked based on the trends observed in the epidemiological curve."
10. Paragraph Structure:
Logical Flow:
Ensure that paragraphs follow a logical sequence. In some parts of the manuscript, ideas are presented in a disjointed manner. For instance, moving from discussing the global spread of COVID-19 directly to the local response in Piauí without a clear transition can be confusing. Consider using transitional phrases to guide the reader.
Author Response
Author's Reply to the Review Report (Reviewer 1)
Comment 1: The manuscript asserts a strong and direct impact of the coping strategies on COVID-19 outcomes in Piauí. However, this conclusion appears to be based primarily on descriptive analysis without sufficient statistical rigor to establish causality. I recommend revisiting these claims and conducting more robust statistical analyses, such as multivariate regression or time-series analysis, to accurately assess the impact of the interventions. This would help avoid overestimating the effectiveness of the strategies without considering other influencing factors. Response 1: [This is a descriptive and documental study, with a qualitative approach, with the objective of analyzing the influence of COVID-19 pandemic strategies implemented by the Public Health Network in the state of Piauí, Brazil, on the outcomes of cases and deaths. The processing of the data contained in the Normative Acts of the Health Department of Piauí - SESAPI was carried out using the IRAMUTEQ® (Interface de R pour les Analyses Multidimensionnelles de Textes et de Questionnaires) software and analyzed by the Descending Hierarchical Classification (DHC), which consists of removing the essence of the text and presenting it in the form of related classes (DENDOGRAMA). The discussion about the correlation between the number of cases and deaths, as a causal relationship, was not characterized as the object of the study, so it was not substantiated. The number of cases and deaths were mentioned to better analyze the dimension of the problem. As a qualitative research, the statistical tests were not deepened. This was highlighted as a limitation of the study, clarifying that the observed relationships require further investigations to confirm causality].
Comment 2: The manuscript could benefit from a more comparative approach, perhaps by comparing Piauí’s strategies and outcomes with those of other Brazilian states or countries with similar socioeconomic conditions. This would provide context and allow for a more nuanced understanding of the effectiveness of the strategies implemented. Response 2: In some paragraphs of the text, the data referring to the State of Piauí were compared with the data from the States of Bahia and Maranhão, which are located in the northeast region and have the same socioeconomic conditions. In Brazil, no documentary analysis studies on the COVID-19 pandemic were located until the preparation of this manuscript, considering Official documents of the Federal Government – Normative Acts.
Comment 3 While the descriptive nature of the study is valuable, the manuscript could be strengthened by incorporating more analytical elements. For instance, the impact of specific strategies on COVID-19 outcomes (cases, mortality, etc.) could be statistically analyzed to draw more definitive conclusions. Response 3: The study has a qualitative approach to better study the subjectivity of facing the pandemic, through the official documents of the state government – Normative Acts.
Comment 4: Correlation Misinterpretation: The discussion on the correlation between the number of cases and deaths suggests a causal relationship that may not be fully substantiated. Correlation does not imply causation, and without controlling for potential confounding variables (e.g., healthcare capacity, public health interventions beyond those documented, population behavior), this interpretation could be misleading. I advise a more cautious presentation of these findings, perhaps by highlighting the limitations of correlation analysis and clarifying that the observed relationships require further investigation to confirm causality. Underestimation of Confounding Factors: The manuscript does not fully account for other factors that may have influenced the COVID-19 outcomes in Piauí, such as the role of healthcare infrastructure, public adherence to guidelines, and the timing and extent of vaccination efforts. These factors are crucial for a comprehensive analysis and should be incorporated into your discussion. Acknowledging these elements will provide a more balanced and accurate portrayal of the situation, thereby enhancing the validity of your conclusions. Response 4: Suggestion heeded in the discussion.
Comment 5: Limited Discussion of Nursing’s Role: Although the title of the manuscript suggests a focus on nursing contributions, the actual discussion of how nursing professionals specifically influenced the outcomes is somewhat limited. To align the content with the title, I recommend expanding this section to include detailed examples of nursing interventions and their impact. Highlighting the unique contributions of nursing, including leadership, patient care innovations, and collaboration with other healthcare sectors, would greatly enhance the manuscript’s relevance and depth. Response 5: Suggestion heeded in the discussion. According to data from the Federal Council of Nursing – COFEN, Nursing played a crucial role during the COVID-19 pandemic in Brazil. Nurses have been on the front lines, performing a variety of essential tasks, from administering vaccines to intensive care of hospitalized patients. In addition to their technical responsibilities, nurses have also been instrumental in humanizing care, serving as a vital link between patients and their families, especially during periods of isolation. They helped mediate communication and provide emotional support, which was crucial for the patients' recovery. Despite these difficulties, such as the lack of Personal Protective Equipment (PPE) and the overload of work, the dedication and commitment of nursing professionals were essential to combat the pandemic and to maintain health services.
Comment 6: The conclusions drawn in the manuscript tend to be general and could benefit from being more specific and actionable. I recommend offering detailed recommendations based on your findings, particularly regarding the ongoing role of nursing professionals in pandemic management and specific strategies for improving public health responses in similar contexts. Response 6: Suggestion heeded at the conclusion.
Comment 7: The study relies heavily on official documents and normative acts, which may introduce bias if these documents do not fully capture the on-the-ground realities or if certain actions were not adequately documented. A discussion on the potential limitations of using these sources would be beneficial. Response 7: Suggestion answered at the end of the discussion.
Comment 8: The references included in the manuscript primarily consist of governmental and official documents. While these sources are certainly valuable, the manuscript would benefit from the inclusion of more peer-reviewed academic articles, especially from journals in public health, epidemiology, and nursing. These sources would provide a stronger foundation for your arguments and contribute to the academic rigor of your study. Response 8: Suggestion heeded in the discussion.
Comment 9: Comments on the Quality of English Language: The manuscript contains several run-on sentences that combine multiple ideas without proper punctuation, making them difficult to follow. For example: "The COVID-19 pandemic determined the adoption of rapid, complex and changeable health action measures, allowing care planning and emergency management strategies to meet the needs of the population. "This sentence could be split into two sentences for clarity: "The COVID-19 pandemic necessitated the adoption of rapid, complex, and changeable health action measures. These measures allowed for effective care planning and the implementation of emergency management strategies to meet the population's needs." Response 9: Suggestion heeded in the sense of making a rigorous translation into English.
Comment 10: Fragmented Sentences: Some sentences lack a clear subject or verb, making them incomplete. For instance: "Considering the above, the study aims to analyze the influence of strategies to fight the COVID-19 pandemic of the Public Health Network of the state of Piauí, Brazil, on the outcome of cases and deaths." This sentence is long and convoluted. It could be restructured as: "This study aims to analyze the influence of COVID-19 pandemic strategies implemented by the Public Health Network in the state of Piauí, Brazil, on case and death outcomes." Response 10: Suggestion met in the objective of the study.
Comment 11: Mixed Tenses: The manuscript frequently switches between past and present tenses, which can confuse the reader. For example: "The data showed more than 481,756,671 cases of COVID-19 worldwide and more than 6,127,981 deaths, with more than 150,432,900 people infected in the Americas up to March 28, 2022." It would be more consistent to use the past tense throughout: "The data showed more than 481,756,671 cases of COVID-19 worldwide, resulting in over 6,127,981 deaths, and more than 150,432,900 infections in the Americas as of March 28, 2022." Response 11: Suggestion heeded in the sense of making a rigorous translation into English.
Comment 12: Singular vs. Plural Mismatch: There are instances where the subject and verb do not agree in number. For example: "The data on COVID-19 cases was collected from SESAPI's information and statistics system." "Data" is plural, so the correct form should be: "The data on COVID-19 cases were collected from SESAPI's information and statistics system." Response 12: Suggestion heeded in the sense of making a rigorous translation into English.
Comment 13: Missing Articles: Articles such as "the," "a," and "an" are often missing, leading to awkward phrasing. For example: "COVID-19 pandemic determined adoption of rapid, complex and changeable health action measures." Should be: "The COVID-19 pandemic determined the adoption of rapid, complex, and changeable health action measures." Response 13: Suggestion heeded in the sense of making a rigorous translation into English.
Comment 14: Improper Word Choice: In some cases, the wrong word is chosen, affecting the clarity of the sentence. For instance: "The actions determined in the normative acts were valid for the emergency that the world was facing." A more appropriate word choice could be: "The measures outlined in the normative acts were appropriate for the emergency the world was facing." Response 14: Suggestion heeded in the sense of making a rigorous translation into English.
Comment 15: Redundant phrases can make sentences wordy and less impactful. For example: "Due to the lack of protective equipment, resulting in unstable or even unsafe patient care, due to the shortage of nursing personnel." This can be streamlined to: "The lack of protective equipment resulted in unstable and unsafe patient care due to the nursing personnel shortage." Response 15: Suggestion heeded in the sense of making a rigorous translation into English.
Comment 16: Commas are often misplaced or missing, which can alter the meaning of a sentence. For example: "The role of nursing professionals through the care provided to the population is the subject of reflection and analysis indicating the need to deepen the understanding." It should be: "The role of nursing professionals, through the care provided to the population, is the subject of reflection and analysis, indicating the need to deepen the understanding." Response 16: Suggestion heeded in the sense of making a rigorous translation into English.
Comment 17: Prepositions are occasionally misused, leading to awkward phrasing. For example: "Increased transmissibility and virulence, associated with changes in the clinical presentation of the disease, low diagnostic efficacies and public health measures, as well as low availability of vaccines and therapies or even mutations in the virus itself led to the emergence of SARS-CoV-2 variants." A clearer phrasing would be: "Increased transmissibility and virulence, combined with changes in the clinical presentation of the disease, low diagnostic efficacy, public health measures, and the limited availability of vaccines and therapies, or even mutations in the virus itself, led to the emergence of SARS-CoV-2 variants." Response 17: Suggestion heeded in the sense of making a rigorous translation into English.
Comment 18: Typographical Errors: There are a few spelling errors or typos that need correction. For instance: "Public Health Netwok" should be corrected to "Public Health Network." Response 18: Suggestion heeded in the sense of making a rigorous translation into English.
Comment 19: Some sentences are ambiguous and need to be rephrased for clarity. For example: "The strategies adopted in Piauí were essential to save lives, with other actions being approved or blocked depending on the behavior of the epidemiological curve." This could be clearer as: "The strategies adopted in Piauí were essential in saving lives. Other actions were either approved or blocked based on the trends observed in the epidemiological curve." Response 19: Suggestion heeded in the sense of making a rigorous translation into English.
Comment 20: Logical Flow: Ensure that paragraphs follow a logical sequence. In some parts of the manuscript, ideas are presented in a disjointed manner. For instance, moving from discussing the global spread of COVID-19 directly to the local response in Piauí without a clear transition can be confusing. Consider using transitional phrases to guide the reader. Response 20: Suggestion answered and revisio

Reviewer 2 Report
Comments and Suggestions for Authors
Dear authors, your manuscript has the potential to significantly contribute to the advancement of knowledge on COVID-19 and the strategies employed to combat it. However, it needs major revisions before addressing other specific aspects. Here is a list of potential issues and areas of concern that you might consider addressing:
1. Title.
It is not clear. Please riformulate it, indicating the design adopted effectively.
2. Structure and organization.
- Please, use paragraph and subparagraph to improve the readability and understanding of the manuscript.
- The introduction might be too detailed in terms of background information, some of which may not be directly relevant to the study's focus on Piauí.
- The manuscript is quite lengthy, and there is a need for better organization. Some sections, like the methods and results, are extensive and may benefit from more concise summarization.
2. Clarity and language.
- Generally, the language is often complex, with long and convoluted sentences that can confuse readers. I recommend a thorough revision in scientific English, focusing on simplifying the text, eliminating unnecessary words and repetitions, and breaking down complex sentences into shorter, more direct ones to enhance clarity and readability.
3. Methodology.
- The methodology section is very detailed, which is good, but it may overwhelm the reader. It might be beneficial to simplify the explanation of statistical methods and the use of specific software like IRAMUTEQ.
- You should clarify the sampling methods and the rationale behind choosing specific normative acts for analysis.
4. Data presentation.
- The presentation of data, particularly in the form of tables and figures, might be too dense. Consider breaking up complex tables into smaller parts or summarizing key points.
- Some of the graphs (e.g., linear regression models) might require better explanation regarding their significance and how they support the study's conclusions. These graphs are difficult to understand.
5. Results interpretation.
- The results are descriptive but lack depth in the interpretation of what these results mean in the broader context of public health in Piauí or Brazil.
- There is a need for clearer links between the data presented and the conclusions drawn. How exactly do the normative acts correlate with changes in COVID-19 cases and deaths?
6. Discussion.
- The discussion section tends to reiterate results without offering new insights or linking the findings to broader research or policy implications.
- More emphasis should be placed on the limitations of the study, particularly concerning the data sources and potential biases in the collection of normative acts.
7. Conclusions.
- The conclusion is somewhat broad and does not fully encapsulate the specific findings of the study. It might benefit from being more focused on the key takeaways and their implications for public health policy.
8. References.
- Ensure all references are relevant. Verify that all citations are correctly formatted and that the citation style is consistent throughout the manuscript.
- A major review is needed (formatting issues, inconsistencies in citation style, accuracy of citations). Several references are URLs to government decrees and other online documents. It is essential to ensure that these links are still active and lead to the correct documents. If any links are broken, this could undermine the validity of the references. Some references are in Portuguese, which is appropriate given the regional focus. However, this may limit accessibility to an international audience if they cannot read Portuguese. Some suggestions: review and standardize formatting, check accessibility, update or replace broken links.
9. Ethical considerations.
- While the manuscript mentions ethical considerations, it could provide more detail on how participant data privacy was ensured, especially since the study involves sensitive public health information. Please use a dedicated paragraph in the Materials and Methods section.
10. Redundancy.
- Some parts of the manuscript are repetitive, particularly in the discussion and methods sections. Reducing redundancy can help streamline the manuscript.
Comments on the Quality of English LanguageA major revision is needed to simplify sentences, correct grammar and punctuation, reduce redundancy, standardize terminology, and ensure the correct use of tense.
Author Response
Comments 1. Title. It is not clear. Please riformulate it, indicating the design adopted effectively. Response 1: Reformulated title: STRATEGIES TO COPE WITH THE COVID-19 PANDEMIC IN THE STATE OF PIAUÍ – BRAZIL: contributions to Nursing.
Comments 2. Structure and organization. - Please, use paragraph and subparagraph to improve the readability and understanding of the manuscript. - The introduction might be too detailed in terms of background information, some of which may not be directly relevant to the study's focus on Piauí. - The manuscript is quite lengthy, and there is a need for better organization. Some sections, like the methods and results, are extensive and may benefit from more concise summarization. Response 2: The manuscript has been summarized and better organized.
Comments 3. Clarity and language. - Generally, the language is often complex, with long and convoluted sentences that can confuse readers. I recommend a thorough revision in scientific English, focusing on simplifying the text, eliminating unnecessary words and repetitions, and breaking down complex sentences into shorter, more direct ones to enhance clarity and readability. Response 3: Suggestion heeded in the sense of making a rigorous translation into English.
Comments 4. Methodology. - The methodology section is very detailed, which is good, but it may overwhelm the reader. It might be beneficial to simplify the explanation of statistical methods and the use of specific software like IRAMUTEQ. Response 4: The explanation of the methods and the use of IRAMUTEQ software have been simplified.
Comments 5- You should clarify the sampling methods and the rationale behind choosing specific normative acts for analysis. Response 5: As the study has a qualitative approach, all Normative Acts published in the State of Piauí were sources of research data.
Comments 6. Data presentation. - The presentation of data, particularly in the form of tables and figures, might be too dense. Consider breaking up complex tables into smaller parts or summarizing key points. - Some of the graphs (e.g., linear regression models) might require better explanation regarding their significance and how they support the study's conclusions. These graphs are difficult to understand. Response 6: The suggestion to better clarify the graphs and especially the Dendrogram with the Classes has been taken into account.
Comments 7. Results interpretation. - The results are descriptive but lack depth in the interpretation of what these results mean in the broader context of public health in Piauí or Brazil. - There is a need for clearer links between the data presented and the conclusions drawn. How exactly do the normative acts correlate with changes in COVID-19 cases and deaths? Response 7: The Normative Acts, which defined the technical-operational protocols for the prevention and control of COVID-19 in the State of Piauí, were directly related to social distancing strategies and the use of personal protective equipment by the population.
Comments 8. Discussion. - The discussion section tends to reiterate results without offering new insights or linking the findings to broader research or policy implications. - More emphasis should be placed on the limitations of the study, particularly concerning the data sources and potential biases in the collection of normative acts. Response 8: Heeded the suggestion linking the findings to policy implications. The suggestion to give more emphasis to the limitations of the study at the end of the introduction was taken into account.
Comments 9. Conclusions. - The conclusion is somewhat broad and does not fully encapsulate the specific findings of the study. It might benefit from being more focused on the key takeaways and their implications for public health policy. Response 9: The suggestion was taken into account – the conclusion focused on the main findings and their implications for public health policy.
Comments 10. References. - Ensure all references are relevant. Verify that all citations are correctly formatted and that the citation style is consistent throughout the manuscript. - A major review is needed (formatting issues, inconsistencies in citation style, accuracy of citations). Several references are URLs to government decrees and other online documents. It is essential to ensure that these links are still active and lead to the correct documents. If any links are broken, this could undermine the validity of the references. Some references are in Portuguese, which is appropriate given the regional focus. However, this may limit accessibility to an international audience if they cannot read Portuguese. Some suggestions: review and standardize formatting, check accessibility, update or replace broken links. Response 10: Made the review suggested in the references.
Comments 11. Ethical considerations. - While the manuscript mentions ethical considerations, it could provide more detail on how participant data privacy was ensured, especially since the study involves sensitive public health information. Please use a dedicated paragraph in the Materials and Methods section. Response 11: The suggestion was taken into account. A paragraph was written about ethical issues in the Materials and Methods section.
Comments 12. Redundancy. - Some parts of the manuscript are repetitive, particularly in the discussion and methods sections. Reducing redundancy can help streamline the manuscript. Response 12: Following the suggestion.
Comments 13. Comments on the Quality of English Language. A major revision is needed to simplify sentences, correct grammar and punctuation, reduce redundancy, standardize terminology, and ensure the correct use of tense. Response 13: Suggestion met. Complete English review done.
Round 2
Reviewer 1 Report
Comments and Suggestions for Authors
Thank you so much for addressing the comments